# Possible Role of Tauroursodeoxycholic Acid (TUDCA) and Antibiotic Administration in Modulating Human Gut Microbiota in Home Enteral Nutrition Therapy for the Elderly: A Case Report

**DOI:** 10.3390/ijms25137115

**Published:** 2024-06-28

**Authors:** Emanuele Francini, Paolo Orlandoni, Debora Sparvoli, Nikolina Jukic Peladic, Maurizio Cardelli, Rina Recchioni, Stefania Silvi, Vilberto Stocchi, Sabrina Donati Zeppa, Antonio Domenico Procopio, Maria Capalbo, Fabrizia Lattanzio, Fabiola Olivieri, Francesca Marchegiani

**Affiliations:** 1Clinic of Laboratory and Precision Medicine, IRCCS INRCA, 60121 Ancona, Italy; r.recchioni@inrca.it (R.R.); a.d.procopio@univpm.it (A.D.P.); fr.marchegiani@inrca.it (F.M.); 2Clinical Nutrition, IRCCS INRCA, 60127 Ancona, Italy; p.orlandoni@inrca.it (P.O.); d.sparvoli@inrca.it (D.S.); n.jukicpeladic@inrca.it (N.J.P.); 3Advanced Technology Center for Aging Research, IRCCS INRCA, 60121 Ancona, Italy; m.cardelli@inrca.it (M.C.); f.olivieri@univpm.it (F.O.); 4School of Biosciences and Veterinary Medicine, University of Camerino, Via Gentile III da Varano, 62032 Camerino, Italy; stefania.silvi@unicam.it; 5Department of Human Science and Promotion of Quality of Life, San Raffaele Rome Telematic University, 00166 Rome, Italy; vilberto.stocchi@uniroma5.it; 6Department of Biomolecular Sciences, University of Urbino Carlo Bo, 61029 Urbino, Italy; sabrina.zeppa@uniurb.it; 7Laboratory of Experimental Pathology, Department of Clinical and Molecular Sciences, Università Politecnica delle Marche, 60100 Ancona, Italy; 8General Direction, IRCCS INRCA, 60124 Ancona, Italy; m.capalbo@inrca.it; 9Scientific Direction, IRCCS INRCA, 60124 Ancona, Italy; f.lattanzio@inrca.it

**Keywords:** home enteral nutrition (HEN), gut microbiota (GM), tauroursodeoxycholic acid (TUDCA), next generation sequencing (NGS), dysbiosis, antibiotic therapy

## Abstract

Tauroursodeoxycholic acid (TUDCA) increases the influx of primary bile acids into the gut. Results obtained on animal models suggested that Firmicutes and Proteobacteria phyla are more resistant to bile acids in rats. As part of a pilot study investigating the role of probiotics supplementation in elderly people with home enteral nutrition (HEN), a case of a 92-year-old woman with HEN is reported in the present study. She lives in a nursing home and suffers from Alzheimer’s disease (AD); the patient had been prescribed TUDCA for lithiasis cholangitis. The aim of this case report is therefore to investigate whether long-term TUDCA administration may play a role in altering the patient’s gut microbiota (GM) and the impact of an antibiotic therapy on the diversity of microbial species. Using next generation sequencing (NGS) analysis of the bacterial 16S ribosomal RNA (rRNA) gene a dominant shift toward Firmicutes and a remodeling in Proteobacteria abundance was observed in the woman’s gut microbiota. Considering the patient’s age, health status and type of diet, we would have expected to find a GM with a prevalence of Bacteroidetes phylum. This represents the first study investigating the possible TUDCA’s effect on human GM.

## 1. Introduction

In cases where the health status of elderly and very frail patients no longer permits adequate nutrition, clinicians may prescribe artificial nutrition (AN), i.e., enteral (EN) or parenteral nutrition (PN) [1]. The aim of AN is to provide adequate amounts of energy, protein and micronutrients and improve nutritional status; to maintain as much gut function as possible; to maintain or improve quality of life; and to reduce morbidity and mortality [2]. EN is made available through two main access routes, namely the nasogastric tube (NGT) and percutaneous endoscopic gastrostomy (PEG). If the patient’s condition allows, the home environment is suitable, and family members and/or caregivers have been trained to perform the activities associated with home enteral nutrition (HEN), therapy can be provided at home [1]. To avoid possible EN intolerance, a pre-digested oligomeric formula can be used by the clinicians. This formula is well tolerated by patients, but the lack of fibre can alter the structure of the bacteria in the colon, the permeability and the gut transit time [3].

Gut microbiota (GM) is the collection of microorganisms (bacteria, viruses and fungi) that have colonized the intestinal tract. The GM of a healthy adult varies between individuals according to geographical location, diet, age, social status and health; it contains an average of 150 bacterial species, 95% of which belong to the phylum Firmicutes and Bacteroidetes, and the remaining 5% to the phyla Actinobacteria, Proteobacteria, Tenericutes, Verrucomicrobia and Fusobacteria, with a Firmicutes/Bacteroidetes ratio (F/B ratio) of approximately 0.8/1 [4]. In addition, the microbiota is influenced by intestinal pH, bile salts [5] and low intake of fibre [3].

Age-dependent exposures can also directly cause ecological perturbations of the gut microbiota up to the development of dysbiosis. The human host and the gut microbiota are in a state of dynamic equilibrium. However, a healthy microbiome may have a direct impact on longevity but, more importantly, it may influence longevity in good health [6]. It is also known that, in addition to the pathological state, the use of drugs/supplements can modulate the composition of the microbiota [7].

Tauroursodeoxycholic acid (TUDCA) can be used to reduce cholestasis in patients. TUDCA is a hydrophilic conjugated bile acid (BA) derivative that is normally produced endogenously in the human liver from the combination of taurine and ursodesoxycholic acid (UDCA). The supplementation of TUDCA can be used to counteract the formation of gallstones and to treat chronic cholestatic liver disease [8,9]. BA are a component of bile; 95% of BA enter the enterohepatic circulation, a process that involves reabsorption by active transport of bile salts in the distal ileum. After absorption, they enter the portal circulation and are rapidly reabsorbed by hepatocytes and secreted into the bile. A total of 5% of BA escape enterohepatic circulation and reach the colon, where they are chemically transformed by three main microbial pathways: deconjugation, dehydrogenation and dihydroxylation reactions [10]. The deconjugation of the primary bile salts taurine and choline is catalysed by bacterial bile salt hydrolases (BSH). The specific function of BSH has not yet been elucidated, but it is thought that one of its main functions is to deconjugate primary bile salts, making them less toxic to bacteria and allowing them to survive in the gut [5,9,10,11,12,13]. Few studies suggest that TUDCA may activate the farnesoid X receptor (FXR), a ligand-mediated transcription factor that controls lipid metabolism, conjugated bile acid homeostasis, liver inflammation, liver regeneration and fibrosis [14,15]. In addition, the composition of the microbiota may be altered in patients with slowed colonic transit and constipation [16,17]. The aim of this case report was to analyze the influence of TUDCA administration in the composition of GM in a 92-years-old woman who was no longer self-sufficient and was living in a nursing home with Alzheimer’s disease.

## 2. Case Presentation

### 2.1. Clinical Case

This study reports the case of a 92-years-old woman (now called 1C) who was no longer self-sufficient and lived in a nursing home (Zaffiro Casa di Cura, Ancona, Italy). The patient weighed 52 kg, was 148 cm tall, had a BMI of 23.7 and was affected by Alzheimer’s disease. The woman presented with slowed colonic transit time with constipation, possibly caused by bed rest and dietary fibre deficiency [3], which was treated pharmacologically with Movicol. She was part of a pilot parallel two-arm randomised intervention protocol designed to evaluate the effects of probiotic administration on the gut microbiota composition of frail elderly subjects with HEN. In this protocol, patients were randomised to two groups:The intervention group was treated with the probiotic SYNBIO^®^ (Synbiotec Srl, Camerino, Italy), 1 capsule/day of 0.26 gr, with a 1:1 mixture of *Lacticaseibacillus rhamnosus* IMC501^®^ and *Lacticaseibacillus paracasei* IMC502^®^ for 30 days;The control group took only the oligomeric mixture of di- and tri-peptides (Peptisorb^®^, Danone Nutricia, Milan, Italy) for 30 days.

Our case report patient was in the control group, she did not take the probiotic and maintained the diet with Peptisorb for the duration of this study (from 12 November 2018 to 18 December 2018). The woman had been fed with Peptisorb since 10 August 2015. At enrolment (t0) and after 30 days (t1), a faecal sample of approximately 5 g was collected from the patient, placed in tubes for copro-culture with the help of the nurse and/or caregiver, and then frozen at −80 °C until NGS analysis was performed. The woman, who was diagnosed with biliary cholangitis in June 2017, started taking TUDCA. She was still taking TUDCA during this study. In addition, the CRF (Case Report Form) analysis showed that the woman had started antibiotic therapy on 7 July 2018 (Ceftriaxone, Rocefin^®^, 1 mL every 16 h for 7 days). The woman also received Lansoprazole (Lansox^®^), Sodium Levothyroxine (Eutirox^®^), Bisoprolol (Congescor^®^), diuretics such as Furosemide (Lasix^®^) and Canrenone (Luvion^®^), Levetiracetam (Keppra^®^), soluble salt of Lysine Acetylsalicylate (Cardirene^®^), and Enoxaparin (Clexane^®^), in accordance with medical prescription.

### 2.2. Next Generation Sequencing of Gut Microbiota

The QIAamp PowerFecal Pro DNA kit from QIAgen (Qiagen GmbH, Hilden, Germany) was used for the extraction of microbial DNA from faeces (1 specimen at t0 and 1 specimen at t1), according to the manufacturer’s instructions. The DNA integrity was evaluated with Genomic DNA ScreenTape in TapeStation Systems (Agilent Technologies, Santa Clara, CA, USA). The DNA Integrity Number (DIN) was ≥ 8 for both samples. Amplification and next generation sequencing were performed according to manufacturer’s instructions. Briefly, the construction of metagenomic amplicons was made with Ion 16S™ Metagenomics Kit (Thermo Fisher Scientific, Waltham, MA, USA) using 3 ng of microbic DNA. This kit provides different primers designed to target six hypervariable regions (V2, V4, V8 and V3, V6-7, V9) of the 16S rRNA gene. The conditions of amplification were as follows: hold 95° 10 min, 22 cycles (denaturation 95° for 30 s, annealing 58° for 30 s and extension 72° for 20 s) and final extension 72° for 7 min. After amplification, PCR products were purified and end-repaired for barcode ligation. A total of 50 ng of amplicons were used to make the libraries with the Ion Plus Fragment Library Kit™ as per the manufacturer’s protocol. The end-repaired product was ligated to 1 µL Adapter and 1 µL Ion Xpress Barcode for each sample (ThermoFisher Scientific, USA). Assessment of the library fragment size distributions and purity was conducted on Agilent TapeStation Systems with D1000 Screen Tape (Agilent Technologies, Santa Clara, CA, USA) according to the manufacturer’s instructions. Each step was followed by purification using volumes of MagSi-DNA beads (Magtivio, Nuth, The Netherlands) and eluted in 20 µL of 1X Low EDTA TE buffer (10 mM Tris base, 0.1 mM EDTA), pH 8.0. Equal volumes of individual barcoded libraries at a concentration of 100 picomolar (pM) were combined in a pool, then diluted to the final concentration of 40 pM. Template preparation and chip loading were performed with the Ion Chef system according to the Ion 510, Ion 520 and Ion 530 Kit-Chef protocol. Sequencing was performed on the Thermo Ion GeneStudio S5 system (Thermo Fisher Scientific, Waltham, MA, USA) using 400 bp sequence run and 520 chip. Base calling and run demultiplexing were performed by Torrent Suite version 5.18.1 (Thermal Fisher Scientific, Waltham, MA, USA) with default parameters. FileExporter version 5.12.0.0 (23) (Thermo Fisher Scientific, Waltham, MA, USA) was used to generate demultiplexed fastq files for each sample. Torrent Suite v. 5.18.1 Ion Reporter (ver 5.20.2.0) (workflow Metagenomics 16Sw1.1 v. 5.18) was used for the analysis with default parameters. Unaligned binary data files (Binary Alignment Map, BAM) generated by the Ion Torrent Suite were uploaded to Ion Reporter and analyzed using default settings.

### 2.3. Alpha Diversity Analyses

The observed species for our case, corresponding to randomly sampled sequences, are expressed by rarefaction curves in Figure 1. In our sample, a flat trend was reached around 50,000 sequences, indicating that a maximum sequencing level had been reached and the sequencing results were reliable. As reported in Figure 1, rarefaction curves display for our case a value of 153 at t0 and 169 at t1.

Alpha diversity expresses the abundance of microbial species present in the gut. We calculated the Shannon, Sympson and Chao1 indices for Alpha diversity analysis. The Shannon index of microbiota was 3.517 at t0 and 3.725 at t1; the Simpson index was 0.833 at t0 and 0.899 at t1; and finally, the Chao1 index was 29 at t0 and 36 at t1.

### 2.4. Gut Microbiota Analysis in the Phylum, Family, Genus and Species Level

The microbial diversity of our case report at t0 and t1 at the phylum level is reported in Figure 2. At t0 (Figure 2a), 1C showed 46% of Firmicutes, 26% of Proteobacteria, 23% of Bacteroidetes, 4% of Actinobacteria and 1% of Synergistetes. At t1 (Figure 2b), 1C showed 70% of Firmicutes, 24% of Bacteroidetes, 3% of Actinobacteria, 2% of Proteobacteria and 1% of Synergistetes. The Firmicutes/Bacteroidetes (F/B) ratio increased from 2 at t0 to 2.91 at t1.

Community richness at family level was considered. The values of percentages of mapped reads at t0 and t1 are reported in Table 1. The top 10 families (in bold in Table 1) that showed the greatest differences between t0 and t1 in the percentages of mapped reads were as follows: Bacteroidaceae (t0: 8.44; t1: 12.45), Clostridiaceae (t0: 8.94; t1: 10.57), Enterobacteriaceae (t0: 22.93; t1: 0.55), Enterococcaceae (t0: 11.09; t1: 17.03), Erysipelotrichaceae (t0: 2.75; t1: 0.74), Lachnospiraceae (t0: 13.54; t1: 10.77), Lactobacillaceae (t0: 0.26; t1: 5.84), Porphyromonadaceae (t0: 10.82; t1: 8.62) and Ruminococcaceae (t0: 5.8; t1: 13.73). In Figure 3 the top 10 representative families are reported. The most represented family at t0 was Enterobacteriaceae (22.93) and Enterococcaceae (17.03) at t1.

Community richness at genus level was considered. The values of percentages of mapped reads at t0 and t1 are reported in Table 2. For those bacteria that could not be identified at the genus level by NGS sequencing, we grouped them together under the term “*Other*” (as shown in Table 2). To better specify the composition of *Other* we reported the list of these bacteria in Appendix A. The top 10 genus (in bold in Table 2) that showed the greatest difference between t0 and t1 in the percentages of mapped reads were as follows: *Bacteroides* (t0: 8.32; t1: 12.36), *Clostridium* (t0: 6.3; t1: 4.9), *Enorma* (t0: 2.53; t1: 0.91), *Enterococcus* (t0: 11.09; t1: 17.03), *Faecalibacterium* (t0: 0.79; t1: 3.95), *Lactobacillus* (t0: 0.25; t1: 5.56), *Odoribacter* (t0: 0.23; t1: 2.32), *Parabacteroides* (t0: 6.42; t1: 2.4), *Ruminococcus* (t0: 5.59; t1: 2.84) and lastly *Other* (t0: 37.85; t1: 33.30). As shown in Table 2, we observed an increase from t0 to t1 in the *Enterococcus* genus (11.09 to 17.03), counterbalanced by a decrease in *Other* (37.85 to 33.30). Butyrate-producing bacteria such as *Roseburia*, which was minimal at t0 (0.02), disappeared at t1 (0.00), while *Faecalibacterium* increased from t0 to t1 (from 0.79 to 3.95). Secondary butyrate-producing bacteria such as *Coprococcus* (t0: 1.24; t1: 1.16) and *Subdoligranulum* (t0: 0.31; t1: 0.10) are present. Among the bacteria involved in protein metabolism, *Ruminococcus* (t0: 0.53; t1: 0.66), *Christensenella* (t0: 0.15; t1: 0.18) and *Clostridium* (t0: 6.30; t1: 4.90) are also found in our patient (Table 2). *Barnesiella* (t0: 1.24; t1: 1.95), *Butyricimonas* (t0: 0.28; t1: 1.21) and *Odoribacter* (t0: 0.23; t1: 2.32), capable of producing butyrate through protein metabolism were also identified. Also, several Gram-negative bacteria with highly inflammatory lipopolysaccharide (LPS) capacity were detected, such as *Bilophila* (t0: 1.02; t1: 0.84), *Enterobacter* (t0:0.39; t1:0. 00), *Escherichia* (t0: 0.06; t1: 0.00), *Escherichia*-*Shigella* (t0: 0.25; t1: 0.01), *Klebsiella* (t0: 1.28; t1: 0.00) and *Sutterella* (t0: 1.82; t1: 0.50). In Figure 4 the top 10 representative genus are reported.

As the number of species revealed by sequencing was very high, the authors decided to report only those species belonging to the two phyla (i.e., Firmicutes and Proteobacteria) which showed the greatest change between t0 and t1 in the patient (Table 3). As expected, Firmicutes (*n* = 52 species) showed a higher number of species than Proteobacteria (*n* = 7 species).

## 3. Discussion

A 92-years-old woman with Alzheimer’s disease living in a nursing home was studied as case report. The patient participated in a parallel two-arm randomised trial involving a probiotic supplemented group and a control group. This study included all subjects on HEN treatment, and our case report belonged to the control group. The bacterial community richness was represented by 153 and 169 OTUs at t0 and t1, respectively. The Alpha diversity of the microbiota community was also similar at both times. The lack of change in these values is probably due to the short time interval (30 days) and the maintenance of an unchanged diet [18,19]. According to the human microbiome project [4], that considers 150 bacterial species as the normal number for healthy adults, a very similar number was observed in the clinical case reported in the present study. Apparently, this result suggests that patient conditions (HEN and Alzheimer’s) do not affect the number of bacterial species found in the microbiota [4]. Notwithstanding, a study found that the diversity of the microbiota decreased in frail elderly compared to healthy adults [20], and other authors have found a decrease in the gut microbiota diversity in Alzheimer’s patients [21]. Perhaps, as observed in these two studies, this is due to constipation and the slow transit time of the colon, which increases the actual number of microbial species, which would otherwise be lower under normal alvus conditions [16,17]. Both at t0 (2.0) and t1 (2.91), the F/B ratio was very high, indicating a dysbiotic state, despite good alpha diversity. A study reported that in the gut of cholic acid (CA)-fed rats, Firmicutes became dominant at the expense of Bacteroidetes, increasing the F/B ratio. This situation led to dysbiosis and liver inflammation [5].

Four months prior to this study, the patient had received Rocefin antibiotic therapy. Antibiotics are one of the factors that have the greatest impact on the diversity of microbial species [19]. In particular, Rocefin’s active ingredient is ceftriaxone, a beta-lactam antibiotic in the third-generation cephalosporin class. Interestingly, in a study, it was found that the use of beta-lactam antibiotics increased the proportion of Bacteroidetes and decreased the Firmicutes and subsequently the alpha diversity [19]. In general, the prolonged use of antibiotics is the cause of a change in the alpha diversity of microbiota; for up to 6 months after the start of therapy, the richness of the microbiota can still be reduced by up to 25% [22]. In the present case report, there still was a good alpha diversity, probably due to constipation, despite the fact that the patient had taken the antibiotic, was in HEN and was a very frail patient affected by Alzheimer’s disease. Two studies reported that the microbiota of AD patients is characterized by an increase in Bacteroidetes, Proteobacteria and Akkermansia and a decrease in Firmicutes [21,23]. An increase in Bacteroidetes over Firmicutes was also observed with age [20]. Unexpectedly, at t0, we observed in our patient a strong presence of Proteobacteria (26%) and a reduced presence of Bacteroidetes (23%) and of Firmicutes (46%). After 30 days (t1), Bacteroidetes remained unchanged (24%), while Firmicutes increased to 70% and Proteobacteria collapsed to 2% (as shown in Figure 2). From June 2017, the woman had been on TUDCA therapy, following biliary cholangitis; she was still taking this at the time of this study.

Interestingly, it has also been shown that administration of 5g TUDCA/kg lithogenic diet to mice increased the F/B ratio by 3.13-fold [9]. Several authors reported the ability of primary bile acids to alter the structure of the microbiota in mice and rats [5,12,24]. BAs have antimicrobial functions and can regulate the structure of the gut microbiota by preventing overgrowth of bacteria in the gut. Furthermore, the composition of bile acids is influenced by the GM bacterial species [10], such as those with BSH enzymatic activity able to deconjugate primary bile salts. This reaction makes the bile salts less toxic to the bacteria [10]. A metagenomic analysis reported that the bacteria most abundant in the BSH enzyme belong to the phylum Firmicutes [25]. Gram-positive bacteria with higher BSH activity are *Lactobacillus*, *Enterococcus* and *Clostridium* genus [12]. Some Gram-negative bacteria, such as Proteobacteria, tolerate bile better than Gram-positive bacteria [5,11]. It was found that rats that were fed a diet supplemented with 1.25 mmol/kg CA showed a selection for the growth of Firmicutes in the gut composition; conversely, a diet supplemented with 5.0 mmol/kg CA promoted the growth of Proteobacteria in addition to Firmicutes. Finally, Bacteroidetes were inhibited in both conditions [5]. Based on the above and the ability of TUDCA to alter the structure of the patient’s GM [9], we hypothesized a role for TUDCA in increasing the proportion of Firmicutes by increasing the amount of BA in the colon. Thus, TUDCA would have increased Firmicutes and Proteobacteria and decreased Bacteroidetes in the patient’s intestinal tract.

At the family level, a strong decrease from t0 to t1 was observed in the Enterobacteriaceae family (22.93 to 0.55), counterbalanced by Ruminococcaceae (5.8 to 13.73). Since the use of Rocefin increases Bacteroidetes 1.5-fold [19], we believe that the change highlighted in the clinical case patient cannot be attributable to this antibiotic. The increased presence of Firmicutes (including Ruminococcaceae) and Proteobacteria (including Enterobacteriaceae) detected in the present study could be more likely consequential to the prolonged use of TUDCA.

The observed lack of *Bifidobacteria*, belonging to the Actinobacteria phylum, agrees with the study that described a reduction in *Bifidobacteria* in institutionalized elderly patients in respect to younger controls [26]. Interestingly, a reduction in these genera was also observed in AD patients [21]. Also, *Roseburia* and *Faecalibacterium* genus—among the butyrate-producing bacteria—were found with a low percentage. Normally, the presence of these genera in the gut is associated with gut health and short chain fatty acid (SCFA) presence [23,27,28]. In elderly patients, the substitution of primary butyrate-producing bacteria, such as *Faecalibacterium*, *Roseburia* and *Aghatobacter*, with alternate butyrate-producing taxa, such as *Odoribacter*, *Butyricimonas*, *Butyrivibrio* and *Oscillospira* was described [29]. In the present case report *Odoribacter* (t0: 0.23; t1: 1.52) and *Butyricimonas* (t0: 0.3; t1: 0.94) were observed. Together with these, other butyrate-producers such as *Coprococcus* (t0: 1.23 to t1: 1.16) and *Subdoligranulum* (t0: 0.31; t1: 0.1) were also detected [30]. Among the Proteobacteria (Gram-negative), *Enterobacter*, *Escherichia*, *Escherichia*-*Shigella*, *Klebsiella*, *Sutterella* and *Bilophila* were detected in the intestine of the clinical case patient. These taxa have a specific LPS in their outer membrane which can trigger strong pro-inflammatory immune responses in the body [31]. In addition, the patient’s slow colonic transit time, which contributes to increased intestinal permeability [16,17], may have facilitated the passage of LPS into the bloodstream, exacerbating a pre-existing state of frailty. Inflammation has also been correlated with the onset and severity of Alzheimer’s disease [32]. An increased presence of the genus *Bilophila* was found in patients with Alzheimer’s disease compared to healthy subjects [21]. *Bilophila wadsworthia*, a bile-resistant bacillus [12], is also resistant to certain beta-lactam antibiotics [33]. This could explain its presence in the clinical case patient’s faeces after antibiotic therapy with Rocefin.

## 4. Conclusions

Metagenomic sequencing of the clinical case patient’s faeces revealed a dysbiotic state due to an excessive F/B ratio. This imbalance could be the result of the prolonged use of TUDCA, which increased the primary BA in the colon. The increased influx of BA favoured the Firmicutes and Proteobacteria phyla at the expense of the Bacteroidetes phylum. The considerations made in this case report may be useful in patients whose GM shows an increase in Bacteroidetes and a decrease in Firmicutes.

Future studies are needed to better understand the role of TUDCA in modulating the gut microbiota in elderly patients.

### Limitations

Our study has some limitations; firstly, as a case report, we only have data from one patient, and we cannot do any statistical analysis. Also, this study included a very old and frail woman who was treated with many drugs, which may have influenced the results. To validate the efficacy of TUDCA administration in influencing GM composition, we need to extend this study to other patients. Another limitation was due to the lack of depth of the NGS, which was not able to detect all bacterial species (grouped under “others”). This could mask the presence of real species in the GM of the patient’s clinical case.

## Figures and Tables

**Figure 1 ijms-25-07115-f001:**
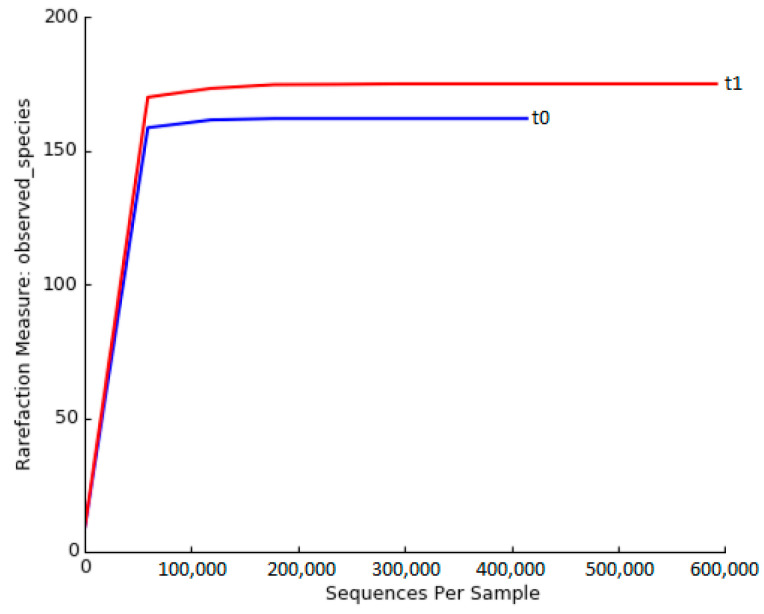
Rarefaction curve of 1C at t0 and t1.

**Figure 2 ijms-25-07115-f002:**
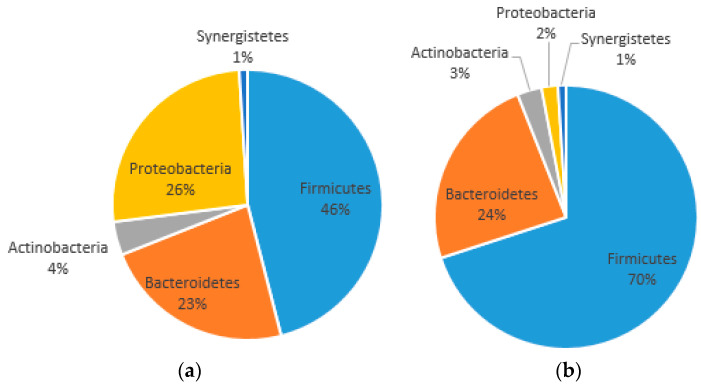
(**a**) Phylum level gut microbiota diversity percentages at t0; (**b**) phylum level gut microbiota diversity percentages at t1.

**Figure 3 ijms-25-07115-f003:**
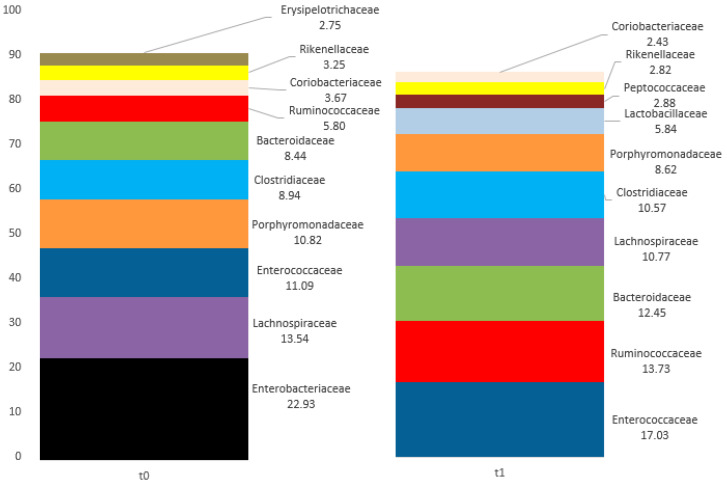
The top 10 families with the highest values detected in 1C at t0 and t1.

**Figure 4 ijms-25-07115-f004:**
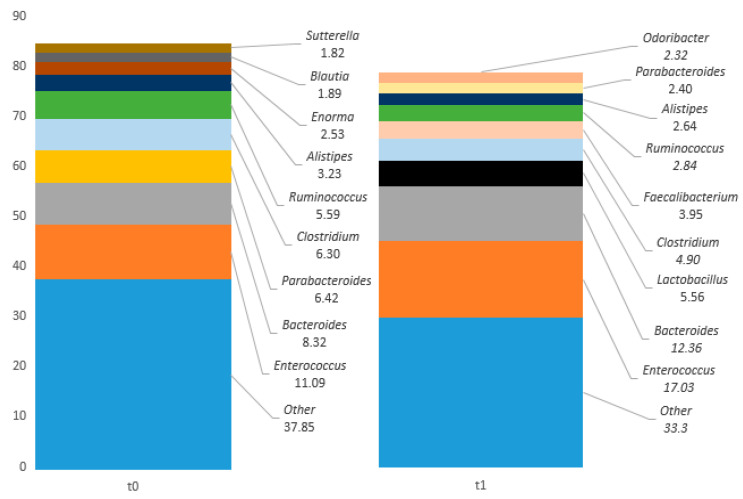
The top 10 genera with the highest values detected in 1C at t0 and t1.

**Table 1 ijms-25-07115-t001:** List of microbial families detected in 1C at t0 and t1.

Phylum	Class	Order	Family	% Mapped Reads t0	% Mapped Reads t1
Firmicutes	Negativicutes	Selenomonadales	Acidaminococcaceae	1.11	0.45
Firmicutes	Bacilli	Bacillales	Bacillaceae	0.00	0.76
**Bacteroidetes**	**Bacteroidia**	**Bacteroidales**	**Bacteroidaceae**	**8.44**	**12.45**
Actinobacteria	Actinobacteria	Bifidobacteriales	Bifidobacteriaceae	0.00	0.08
Firmicutes	Clostridia	Clostridiales	Catabacteriaceae	0.00	0.07
Firmicutes	Clostridia	Clostridiales	Christensenellaceae	0.74	1.92
**Firmicutes**	**Clostridia**	**Clostridiales**	**Clostridiaceae**	**8.94**	**10.57**
Firmicutes	Clostridia	Clostridiales	Clostridiales Family XI. Incertae Sedis	0.01	0.00
Firmicutes	Clostridia	Clostridiales	Clostridiales Family XII. Incertae Sedis	0.06	0.01
Firmicutes	Clostridia	Clostridiales	Clostridiales Family XIII. Incertae Sedis	0.10	0.12
Actinobacteria	Actinobacteria	Coriobacteriales	Coriobacteriaceae	3.67	2.43
Actinobacteria	Actinobacteria	Actinomycetales	Corynebacteriaceae	0.25	0.08
Proteobacteria	Deltaproteobacteria	Desulfovibrionales	Desulfovibrionaceae	1.02	0.84
**Proteobatteri**	**Gammaproteobacteria**	**Enterobacteriales**	**Enterobacteriaceae**	**22.93**	**0.55**
**Firmicutes**	**Bacilli**	**Lactobacillales**	**Enterococcaceae**	**11.09**	**17.03**
**Firmicutes**	**Erysipelotrichi**	**Erysipelotrichaceae**	**Erysipelotrichaceae**	**2.75**	**0.74**
Firmicutes	Clostridia	Clostridiales	Eubacteriaceae	0.50	0.79
Firmicutes	Clostridia	Clostridiales	Gracilibacteraceae	0.07	1.00
**Firmicutes**	**Clostridia**	**Clostridiales**	**Lachnospiraceae**	**13.54**	**10.77**
**Firmicutes**	**Bacilli**	**Lactobacillales**	**Lactobacillaceae**	**0.26**	**5.84**
Firmicutes	Clostridia	Clostridiales	Oscillospiraceae	0.51	1.80
Firmicutes	Bacilli	Bacillales	Paenibacillaceae	0.00	0.06
**Firmicutes**	**Clostridia**	**Clostridiales**	**Peptococcaceae**	**0.48**	**2.88**
**Bacteroidetes**	**Bacteroidia**	**Bacteroidales**	**Porphyromonadaceae**	**10.82**	**8.62**
Proteobacteria	Gammaproteobacteria	Pseudomonadales	Pseudomonadaceae	0.17	0.05
Bacteroidetes	Bacteroidia	Bacteroidales	Rikenellaceae	3.25	2.82
**Firmicutes**	**Clostridia**	**Clostridiales**	**Ruminococcaceae**	**5.80**	**13.73**
Firmicutes	Bacilli	Bacillales	Staphylococcaceae	0.01	0.00
Firmicutes	Bacilli	Lactobacillales	Streptococcaceae	0.04	0.00
Proteobacteria	Betaproteobacteria	Burkholderiales	Sutterellaceae	1.82	0.50
Synergistetes	Syneristia	Synergistales	Synergistaceae	1.28	1.23
Firmicutes	Clostridia	Clostridiales	Thermoanaerobacterales Family III. Incertae Sedis	0.00	1.23
Firmicutes	Clostridia	Clostridiales	Unclassified Clostridiales	0.34	0.48
Lentisphaeria	Lentisphaeria	Victivallales	Victivallaceae	0.01	0.02

**Table 2 ijms-25-07115-t002:** List of microbial genera detected in 1C at t0 and t1.

Phylum	Class	Order	Family	Genus	% Mapper Reads t0	% Mapped Reads t1
Firmicutes	Negativicutes	Selenomonadales	Acidaminococcaceae	*Acidaminococcus*	0.00	0.01
Bacteroidetes	Bacteroidia	Bacteroidales	Rikenellaceae	*Alistipes*	3.23	2.64
-	-	-	-	** *Other ** **	**37.85**	**33.30**
Firmicutes	Clostridia	Clostridiales	Lachnospiraceae	*Anaerostipes*	0.00	0.01
Firmicutes	Clostridia	Clostridiales	Ruminococcaceae	*Anaerotroncus*	0.39	0.30
**Bacteroidetes**	**Bacteroidia**	**Bacteroidales**	**Bacteroidaceae**	** *Bacteroides* **	**8.32**	**12.36**
Bacteroidetes	Bacteroidia	Bacteroidales	Porphyromonadaceae	*Barnesiella*	1.24	1.75
Actinobacteria	Actinobacteria	Bifidobacteriales	Bifidobacteriaceae	*Bifidobacterium*	0.00	0.08
Proteobacteria	Deltaproteobacteria	Desulfovibrionales	Desulfovibrionaceae	*Bilophila*	1.02	0.84
Firmicutes	Clostridia	Clostridiales	Lachnospiraceae	*Blautia*	1.89	0.55
Bacteroidetes	Bacteroidia	Bacteroidales	Porphyromonadaceae	*Butyricimonas*	0.28	1.21
Firmicutes	Clostridia	Clostridiales	Christensenellaceae	*Christensenella*	0.15	0.18
Proteobacteria	Gammaproteobacteria	Enterobacteriales	Enterobacteriaceae	*Citrobacter*	0.00	0.00
Synergistetes	Synergistia	Synergistales	Synergistaceae	*Cloacibacillus*	0.22	0.13
**Firmicutes**	**Clostridia**	**Clostridiales**	**Clostridiaceae**	** *Clostridium* **	**6.30**	**4.90**
Firmicutes	Clostridia	Clostridiales	Lachnospiraceae	*Coprococcus*	1.24	1.16
Actinobacteria	Actinobacteria	Actinomycetales	Corynebacteriaceae	*Corynebacterium*	0.25	0.08
Actinobacteria	Actinobacteria	Coriobacteriales	Coriobacteriaceae	*Denitrobacterium*	0.03	0.23
Firmicutes	Clostridia	Clostridiales	Lachnospiraceae	*Dorea*	0.02	0.77
Actinobacteria	Actinobacteria	Coriobacteriales	Coriobacteriaceae	*Eggerthella*	0.97	0.8
Actinobacteria	Actinobacteria	Coriobacteriales	Coriobacteriaceae	*Enorma*	2.53	0.91
Proteobacteria	Gammaproteobacteria	Enterobacteriales	Enterobacteriaceae	*Enterobacter*	0.39	0.00
**Firmicutes**	**Bacilli**	**Lactobacillales**	**Enterococcaceae**	** *Enterococcus* **	**11.09**	**17.03**
Proteobacteria	Gammaproteobacteria	Enterobacteriales	Enterobacteriaceae	*Escherichia*	0.06	0.00
Proteobacteria	Gammaproteobacteria	Enterobacteriales	Enterobacteriaceae	*Escherichia/Shigella*	0.25	0.01
Firmicutes	Clostridia	Clostridiales	Eubacteriaceae	*Eubacterium*	0.37	0.16
**Firmicutes**	**Clostridia**	**Clostridiales**	**Ruminococcaceae**	** *Faecalibacterium* **	**0.79**	**3.95**
Firmicutes	Erysipelotrichia	Erysipelotrichales	Erysipelotrichaceae	*Faecalicoccus*	0.97	0.26
Firmicutes	Clostridia	Clostridiales	Clostridiales Family XI. Incertae Sedis	*Finegoldia*	0.01	0.00
Firmicutes	Clostridia	Clostridiales	Unclassified Clostridiales	*Flavonifractor*	0.08	0.01
Actinobacteria	Actinobacteria	Coriobacteriales	Coriobacteriaceae	*Gordonibacter*	0.07	0.4
Firmicutes	Erysipelotrichia	Erysipelotrichales	Erysipelotrichaceae	*Holdemania*	0.16	0.06
Proteobacteria	Gammaproteobacteria	Enterobacteriales	Enterobacteriaceae	*Klebsiella*	1.28	0.00
Firmicutes	Clostridia	Clostridiales	Lachnospiraceae	*Lachnoclostridium*	0.29	0.04
**Firmicutes**	**Bacilli**	**Lactobacillales**	**Lactobacillaceae**	** *Lactobacillus* **	**0.25**	**5.56**
Proteobacteria	Gammaproteobacteria	Enterobacteriales	Enterobacteriaceae	*Morganella*	0.10	0.10
**Bacteroidetes**	**Bacteroidia**	**Bacteroidales**	**Porphyromonadaceae**	** *Odoribacter* **	**0.23**	**2.32**
**Bacteroidetes**	**Bacteroidia**	**Bacteroidales**	**Porphyromonadaceae**	** *Parabacteroides* **	**6.42**	**2.40**
Actinobacteria	Actinobacteria	Coriobacteriales	Coriobacteriaceae	*Paraeggerthella*	0.03	0.07
Firmicutes	Negativicutes	Selenomonadales	Acidaminococcaceae	*Phascolarctobacterium*	1.11	0.44
Proteobacteria	Gammaproteobacteria	Enterobacteriales	Enterobacteriaceae	*Proteus*	0.01	0.00
Firmicutes	Clostridia	Clostridiales	Unclassified Clostridiales	*Pseudoflavonifractor*	0.00	0.01
Firmicutes	Clostridia	Clostridiales	Lachnospiraceae	*Roseburia*	0.02	0.00
**Firmicutes**	**Clostridia**	**Clostridiales**	**Lachnospiraceae**	** *Ruminococcus* **	**5.59**	**2.84**
Firmicutes	Clostridia	Clostridiales	Ruminococcaceae	*Ruminococcus*	0.53	0.66
Actinobacteria	Actinobacteria	Coriobacteriales	Coriobacteriaceae	*Senegalimassilia*	0.03	0.01
Proteobacteria	Gammaproteobacteria	Enterobacteriales	Enterobacteriaceae	*Serratia*	0.01	0.00
Firmicutes	Bacilli	Bacillales	Staphylococcaceae	*Staphylococcus*	0.01	0.00
**Firmicutes**	**Erysipelotrichia**	**Erysipelotrichales**	**Erysipelotrichaceae**	** *Streptococcus* **	**1.66**	**0.38**
Firmicutes	Clostridia	Clostridiales	Ruminococcaceae	*Subdoligranulum*	0.31	0.10
**Proteobacteria**	**Betaproteobacteria**	**Burkholderiales**	**Sutterellaceae**	** *Sutterella* **	**1.82**	**0.50**

* Represents the sum of all bacteria that were not identified with a specific genus but only at family level. *Other* represents the sum of all bacteria that were not identified with a specific genus but only at family level.

**Table 3 ijms-25-07115-t003:** List of species belonging to Firmicutes and Proteobacteria detected in 1C.

Phylum	Order	Class	Family	Genus	Species	% Mapped Reads t0	% Mapped Reads t1
Firmicutes	Bacilli	Lactobacillales	Enterococcaceae	*Enterococcus*	*avium*	3.54	6.15
Firmicutes	Bacilli	Lactobacillales	Enterococcaceae	*Enterococcus*	*faecalis*	0.05	1.82
Firmicutes	Bacilli	Lactobacillales	Enterococcaceae	*Enterococcus*	*faecium*	0.00	0.01
Firmicutes	Bacilli	Lactobacillales	Enterococcaceae	*Enterococcus*	*gallinarum*	0.06	0.06
Firmicutes	Bacilli	Lactobacillales	Enterococcaceae	*Enterococcus*	*hermanniensis*	0.00	0.01
Firmicutes	Bacilli	Lactobacillales	Enterococcaceae	*Enterococcus*	*lemanii*	1.10	1.41
Firmicutes	Bacilli	Lactobacillales	Enterococcaceae	*Enterococcus*	*malodoratus*	0.01	0.01
Firmicutes	Bacilli	Lactobacillales	Lactobacillaceae	*Lactobacillus*	*paracasei*	0.09	2.49
Firmicutes	Bacilli	Lactobacillales	Lactobacillaceae	*Lactobacillus*	*zeae*	0.00	0.10
Firmicutes	Clostridia	Clostridiales	Christensenellaceae	*Christensenella*	*minuta*	0.13	0.06
Firmicutes	Clostridia	Clostridiales	Clostridiaceae	*Clostridium*	*aldenense*	0.11	0.05
Firmicutes	Clostridia	Clostridiales	Clostridiaceae	*Clostridium*	*asparagiforme*	0.06	0.14
Firmicutes	Clostridia	Clostridiales	Clostridiaceae	*Clostridium*	*bolteae*	0.08	0.11
Firmicutes	Clostridia	Clostridiales	Clostridiaceae	*Clostridium*	*citroniae*	0.01	0.00
Firmicutes	Clostridia	Clostridiales	Clostridiaceae	*Clostridium*	*clostridioforme*	0.02	0.00
Firmicutes	Clostridia	Clostridiales	Clostridiaceae	*Clostridium*	*glycyrrhizinilyticum*	0.15	0.01
Firmicutes	Clostridia	Clostridiales	Clostridiaceae	*Clostridium*	*hathewayi*	0.16	0.02
Firmicutes	Clostridia	Clostridiales	Clostridiaceae	*Clostridium*	*lavalense*	0.38	0.03
Firmicutes	Clostridia	Clostridiales	Clostridiaceae	*Clostridium*	*scindens*	1.71	1.83
Firmicutes	Clostridia	Clostridiales	Clostridiaceae	*Clostridium*	*symbiosum*	0.08	0.03
Firmicutes	Clostridia	Clostridiales	Clostridiales Family XI. Incertae Sedis	*Finegoldia*	*magna*	0.01	0.00
Firmicutes	Clostridia	Clostridiales	Eubacteriaceae	*Eubacterium*	*callanderi*	0.04	0.01
Firmicutes	Clostridia	Clostridiales	Eubacteriaceae	*Eubacterium*	*contortum*	0.01	0.01
Firmicutes	Clostridia	Clostridiales	Eubacteriaceae	*Eubacterium*	*limosum*	0.07	0.03
Firmicutes	Clostridia	Clostridiales	Lachnospiraceae	*Anaerostipes*	*caccae*	0.00	0.01
Firmicutes	Clostridia	Clostridiales	Lachnospiraceae	*Blautia*	*faecis*	0.02	0.00
Firmicutes	Clostridia	Clostridiales	Lachnospiraceae	*Blautia*	*coccoides*	0.02	0.03
Firmicutes	Clostridia	Clostridiales	Lachnospiraceae	*Blautia*	*hansenii*	0.06	0.03
Firmicutes	Clostridia	Clostridiales	Lachnospiraceae	*Blautia*	*hydrogenotrophica*	0.00	0.24
Firmicutes	Clostridia	Clostridiales	Lachnospiraceae	*Blautia*	*producta*	0.03	0.07
Firmicutes	Clostridia	Clostridiales	Lachnospiraceae	*Blautia*	*wexlerae*	1.02	0.07
Firmicutes	Clostridia	Clostridiales	Lachnospiraceae	*Coprococcus*	*comes*	1.24	1.16
Firmicutes	Clostridia	Clostridiales	Lachnospiraceae	*Dorea*	*dorea*	0.00	0.68
Firmicutes	Clostridia	Clostridiales	Lachnospiraceae	*Dorea*	*longicatena*	0.00	0.01
Firmicutes	Clostridia	Clostridiales	Lachnospiraceae	*Lachnoclostridium*	*clostridioforme*	0.01	0.00
Firmicutes	Clostridia	Clostridiales	Lachnospiraceae	*Lachnoclostridium*	*lavalense*	0.09	0.01
Firmicutes	Clostridia	Clostridiales	Lachnospiraceae	*Roseburia*	*faecis*	0.01	0.00
Firmicutes	Clostridia	Clostridiales	Lachnospiraceae	*Roseburia*	*hominis*	0.01	0.00
Firmicutes	Clostridia	Clostridiales	Lachnospiraceae	*Ruminococcus*	*gnavus*	1.95	0.29
Firmicutes	Clostridia	Clostridiales	Lachnospiraceae	*Ruminococcus*	*torques*	2.97	1.86
Firmicutes	Clostridia	Clostridiales	Ruminococcaceae	*Anaerotruncus*	*colihominis*	0.10	0.04
Firmicutes	Clostridia	Clostridiales	Ruminococcaceae	*Faecalibacterium*	*prausnitzii*	0.57	3.37
Firmicutes	Clostridia	Clostridiales	Ruminococcaceae	*Ruminococcus*	*faecis*	0.09	0.01
Firmicutes	Clostridia	Clostridiales	Ruminococcaceae	*Ruminococcus*	*gauvreauii*	0.00	0.01
Firmicutes	Clostridia	Clostridiales	Unclassified Clostridiales	*Flavonifractor*	*plautii*	0.08	0.01
Firmicutes	Erysipelotrichia	Erysipelotrichales	Erysipelotrichaceae	*Eubacterium*	*dolichum*	0.00	0.02
Firmicutes	Erysipelotrichia	Erysipelotrichales	Erysipelotrichaceae	*Clostridium*	*innocuum*	0.00	0.02
Firmicutes	Erysipelotrichia	Erysipelotrichales	Erysipelotrichaceae	*Faecalicoccus*	*pleomorphus*	0.93	0.25
Firmicutes	Erysipelotrichia	Erysipelotrichales	Erysipelotrichaceae	*Holdemania*	*filiformis*	0.13	0.04
Firmicutes	Erysipelotrichia	Erysipelotrichales	Erysipelotrichaceae	*Streptococcus*	*pleomorphus*	1.62	0.38
Firmicutes	Negativicutes	Selenomonadales	Acidaminococcaceae	*Acidaminococcus*	*intestini*	0.00	0.01
Firmicutes	Negativicutes	Selenomonadales	Acidaminococcaceae	*Phascolarctobacterium*	*faecium*	1.11	0.44
Proteobacteria	Deltaproteobacteria	Desulfovibrionales	Desulfovibrionaceae	*Bilophila*	*wadsworthia*	0.75	0.64
Proteobacteria	Gammaproteobacteria	Enterobacteriales	Enterobacteriaceae	*Enterobacter*	*sacchari*	0.39	0.00
Proteobacteria	Gammaproteobacteria	Enterobacteriales	Enterobacteriaceae	*Escherichia*	*coli*	0.06	0.00
Proteobacteria	Gammaproteobacteria	Enterobacteriales	Enterobacteriaceae	*Escherichia/Shigella*	*coli*	0.25	0.01
Proteobacteria	Gammaproteobacteria	Enterobacteriales	Enterobacteriaceae	*Klebsiella*	*pneumoniae*	0.12	0.00
Proteobacteria	Gammaproteobacteria	Enterobacteriales	Enterobacteriaceae	*Klebsiella*	*variicola*	0.23	0.00
Proteobacteria	Gammaproteobacteria	Enterobacteriales	Enterobacteriaceae	*Morganella*	*morganii*	0.10	0.10

## Data Availability

Data are contained within the article.

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
