# Peer review of "Possible Role of Tauroursodeoxycholic Acid (TUDCA) and Antibiotic Administration in Modulating Human Gut Microbiota in Home Enteral Nutrition Therapy for the Elderly: A Case Report"

_ijms, 2024, doi:10.3390/ijms25137115_

Round 1

Reviewer 1 Report

Comments and Suggestions for Authors

The reviewed case report describes the putative effect of tauroursodeoxycholic acid on the composition of the intestinal flora in an elderly woman additionally suffering from Alzheimer's disease. Bacterial composition was analyzed by metagenomic sequencing of the patient's faeces. In general, it is interesting work with some clinical importance.

My concerns:

Title - authors try to determine the "role" of TUDCA. But how can the role be determined using a single case and without a reference point? From the description of the methodology, it appears that samples were taken once at t0 and t1.

Line 109 - please clarify what time passed from t0 to t1.

Line 232 - please correct minor language errors like "raprestned"

General - please avoid using citation in the form of a name. Please use the journal guidelines.

Author Response

 Response to Reviewer 1

1. Summary

Thank you very much for taking the time to review this manuscript. Please find the detailed responses below and the corresponding revisions/corrections highlighted/in track changes in the re-submitted files. We think our manuscript is better now, thanks to the valuable suggestions.

2. Questions for General Evaluation

Reviewer’s Evaluation

Response and Revisions

Does the introduction provide sufficient background and include all relevant references?

Can be improved

We have tried to improve following the reviewer’s suggestions.

Is the research design appropriate?

Must be improved

We have tried to improve following the reviewer’s suggestions

Are the methods adequately described?

Must be improved

We have tried to improve following the reviewer’s suggestions.

Are the results clearly presented?

Must be improved

We have tried to improve following the reviewer’s suggestions.

Are the conclusions supported by the results?

Can be improved

We have tried to improve following the reviewer’s suggestions.

3. Point-by-point response to Comments and Suggestions for Authors

The reviewed case report describes the putative effect of tauroursodeoxycholic acid on the composition of the intestinal flora in an elderly woman additionally suffering from Alzheimer's disease. Bacterial composition was analyzed by metagenomic sequencing of the patient's faeces. In general, it is interesting work with some clinical importance. The authors address a highly relevant topic for physicians in everyday clinical practice.

Comments and suggestions:

Title

- Authors try to determine the "role" of TUDCA. But how can the role be determined using a single case and without a reference point? From the description of the methodology, it appears that samples were taken once at t0 and t1.

Response: We thank the reviewer for his comment. According with the reviewer’s suggestion, we have changed the title of the case report by adding “Possible” before “role”. Regarding the reference point, nothing is currently known about the composition of GM in humans. The available data are only from mice and rats (Saiful, I.K.B.M.; Islam, K.B.; Fukiya, S.; Hagio, M.; Fujii, N.; Ishizuka, S.; Ooka, T.; Ogura, Y.; Hayashi, T.; Yokota, A. Bile acid is a host factor that regulates the composition of the cecal microbiota in rats. Gastroenterology. 2011, 141, 1773-81. doi: 10.1053/j.gastro.2011.07.046; Lu, Q.; Jiang, Z.; Wang, Q.; Hu, H.; Zhao, G. The effect of Tauroursodeoxycholic acid (TUDCA) and gut microbiota on murine gallbladder stone formation. Ann Hepatol. 2021, 23, 100289. doi: 10.1016/j.aohep.2020.100289).

-          Lines 109: please clarify what time passed from t0 to t1

Response: We thank the reviewer for his comment and we have clarified that there were 30 days between T0 and T1 (line 109: highlighted in red)

-          Line 232 - please correct minor language errors like "raprestned"

Response: We thank the reviewer for the comment. We have corrected it (line 232: highlighted in red).

-          General - please avoid using citation in the form of a name. Please use the journal guidelines.

Response: We thank the reviewer for his comment. As suggested, we have avoided using the author's name in the text (lines 239, 241, 242, 245, 252, 259, 268, 275, 279, 288, 289, 295, and 313: highlighted in red).

Reviewer 2 Report

Comments and Suggestions for Authors

The authors present an interesting study in which the profile of the gut microbiota is examined with respect to the administration of tauroursodeoxycholic acid (TUDCA). TUDCA is routinely prescribed to reduce the risk of bile acid related complications, and some clinical populations receive the compound for long courses owing to their health-related needs. In this instance, the authors aim to examine the impact of long-term TUDCA administration on the gut microbiota of an elderly patient in receipt of such. Using next gen sequencing and other profiling methods, the authors suggest a great shift in the numbers of certain families of bacteria, notably the Firmicutes and Proteobacteria, which is a stark contrast to that which was expected. As such, this study highlights an unexpected finding of this therapy, which is of significant importance given the growing importance and recognition of the gut microbiota on overall health.

In reviewing the manuscript, I made a couple of observations. The authors should consider the following when preparing a suitable revision. 

1.      The tables (1 and 2) require reformatting as some labels are spliced between different rows i.e. ‘Gammaproeob’ and ‘acteria’, and some positioning of text could be centred, etc. Table 3 is improved over 1 and 2, but also requires some minor edits to have it at publishable standard.

2.      How long were the samples stored between collection and NGS being performed? Was the sample stored in any way that was designed to preserve the integrity of the sample beyond freezing it?

3.      The formatting of Figure 1 could be improved slightly to make the labels easier to read.

Author Response

Response to Reviewer 2

1. Summary

Thank you very much for taking the time to review this manuscript. Please find the detailed responses below and the corresponding revisions/corrections highlighted/in track changes in the re-submitted files. We think our manuscript is better now, thanks to the valuable suggestions.

2. Questions for General Evaluation

Reviewer’s Evaluation

Response and Revisions

Does the introduction provide sufficient background and include all relevant references?

Yes

Is the research design appropriate?

Yes

Are the methods adequately described?

Yes

Are the results clearly presented?

Can be improved

We have tried to improve following the reviewer’s suggestions.

Are the conclusions supported by the results?

Yes

3. Point-by-point response to Comments and Suggestions for Authors

The authors present an interesting study in which the profile of the gut microbiota is examined with respect to the administration of tauroursodeoxycholic acid (TUDCA). TUDCA is routinely prescribed to reduce the risk of bile acid related complications, and some clinical populations receive the compound for long courses owing to their health-related needs. In this instance, the authors aim to examine the impact of long-term TUDCA administration on the gut microbiota of an elderly patient in receipt of such. Using next gen sequencing and other profiling methods, the authors suggest a great shift in the numbers of certain families of bacteria, notably the Firmicutes and Proteobacteria, which is a stark contrast to that which was expected. As such, this study highlights an unexpected finding of this therapy, which is of significant importance given the growing importance and recognition of the gut microbiota on overall health.

In reviewing the manuscript, I made a couple of observations. The authors should consider the following when preparing a suitable revision.

Comments and suggestions:

-          The tables (1 and 2) require reformatting as some labels are spliced between different rows i.e. ‘Gammaproeob’ and ‘acteria’, and some positioning of text could be centred, etc. Table 3 is improved over 1 and 2, but also requires some minor edits to have it at publishable standard.

Response: According with the reviewer’s suggestion we reformatting tables 1, 2, and 3 so that the words fit on a single line (tables 1, 2 and 3: highlighted in red).

-          How long were the samples stored between collection and NGS being performed? Was the sample stored in any way that was designed to preserve the integrity of the sample beyond freezing it?

Response: Samples were collected from the patient on 12 November and 18 December 2018. They were immediately frozen at -80°C until NGS analysis (performed on 30 May 2023).

-          The formatting of Figure 1 could be improved slightly to make the labels easier to read.

Response: We thank a lot the reviewer for the comment. We have improved the figure 1 by making it simpler and clearer.

Reviewer 3 Report

Comments and Suggestions for Authors

The study conducted by Francini et al is well written and presented. However, before its publication, I have minor suggestions to the authors, as follows:

The authors should provide the study's main conclusions at the end of the abstract, as well as some directions for future investigations based on these results. What do we learn from this case report and what should be done in the next steps? Also, at the end of the manuscript, in the Conclusions section, this should be elaborated.

The study limitations should be mentioned at the end of the Discussion section.

Why did the authors opt to investigate a 92-year-old woman, no longer self-sufficient with Alzheimer’s disease? Clarify this in the manuscript.

Author Response

Response to Reviewer 3

1. Summary

Thank you very much for taking the time to review this manuscript. Please find the detailed responses below and the corresponding revisions/corrections highlighted/in track changes in the re-submitted files. We think our manuscript is better now, thanks to the valuable suggestions.

2. Questions for General Evaluation

Reviewer’s Evaluation

Response and Revisions

Does the introduction provide sufficient background and include all relevant references?

Yes

Is the research design appropriate?

Yes

Are the methods adequately described?

Can be improved

We have tried to improve following the reviewer’s suggestions.

Are the results clearly presented?

Yes

Are the conclusions supported by the results?

Can be improved

We have tried to improve following the reviewer’s suggestions.

3. Point-by-point response to Comments and Suggestions for Authors

The study conducted by Francini et al is well written and presented. However, before its publication, I have minor suggestions to the authors, as follows:

Abstract and Conclusions:

-          The authors should provide the study's main conclusions at the end of the abstract, as well as some directions for future investigations based on these results. What do we learn from this case report and what should be done in the next steps? Also, at the end of the manuscript, in the Conclusions section, this should be elaborated

Response: We thank a lot the reviewer for his comment and in order to stay within the word limit of the abstract (200), we have included a final sentence (lines 36 and 37: highlighted in red). About the Conclusions section, we have added a sentence that could indicate future uses of TUDCA (lines 322 and 323: highlighted in red).

-     The study limitations should be mentioned at the end of the Discussion section

Response: According with the reviewer’s suggestion we have added a “Limitations” section at the end of the conclusions paragraph (lines 327 to 334: highlighted in red).

-          Why did the authors opt to investigate a 92-year-old woman, no longer self-sufficient with Alzheimer’s disease? Clarify this in the manuscript.

Response: We decided to analyses this elderly patient, who was fed via HEN as the other patients of the original protocol, because she had a significantly different GM structure than the other cases. The analysis of pharmacotherapy revealed the use of TUDCA since 2017. The current literature supports the probable effect of TUDCA as a modifier of intestinal bacteria.